# Amino acid variants of SARS-CoV-2 papain-like protease have impact on drug binding

**Agata P. Perlinska**[1], **Adam Stasiulewicz**[1,2], **Mai Lan Nguyen**[1,3], **Karolina Swiderska**[4], **Mikolaj Zmudzinski**[4], **Alicja W. Maksymiuk**[1,5], **Marcin Drag**[4], **Joanna I. Sulkowska**[1] *

1 Centre of New Technologies, University of Warsaw, Warsaw, Poland, 2 Department of Drug Chemistry, Faculty of Pharmacy, Medical University of Warsaw, Warsaw, Poland, 3 Faculty of Mathematics, Informatics and Mechanics, University of Warsaw, Warsaw, Poland, 4 Department of Chemical Biology and Bioimaging, Wroclaw University of Science and Technology, Wroclaw, Poland, 5 Department of Genetics, University of Cambridge, Cambridge, United Kingdom

* jsulkowska@cent.uw.edu.pl

**Data Availability Statement:** All relevant data are within the manuscript and its Supporting information files.

## Abstract

The novel severe acute respiratory syndrome coronavirus 2 (SARS-CoV-2) has caused both a health and economic crisis around the world. Its papain-like protease (PLpro) is one of the protein targets utilized in designing new drugs that would aid vaccines in the fight against the virus. Although there are already several potential candidates for a good inhibitor of this protein, the degree of variability of the protein itself is not taken into account. As an RNA virus, SARS-CoV-2 can mutate to a high degree, but PLpro variability has not been studied to date. Based on sequence data available in databases, we analyzed the mutational potential of this protein. We focused on the effect of observed mutations on inhibitors' binding mode and their efficacy as well as protein's activity. Our analysis identifies five mutations that should be monitored and included in the drug design process: P247S, E263D-Y264H and T265A-Y268C.

## Author summary

Antiviral drugs often work by tightly fitting into a viral protein and interfering with its function. However, viruses can evolve at a high rate and even a single amino acid change in the binding pocket can greatly disrupt protein-ligand interactions and decrease drug's efficiency. Currently, one of the most important viruses is the novel severe acute respiratory syndrome coronavirus 2 (SARS-CoV-2) responsible for a global health and economical crisis. Here, we demonstrate how naturally-occurring mutations of SARS-CoV-2 papain-like protease (PLpro) reduce the effectiveness of existing inhibitors. First, we employ theoretical calculations of mutations' impact and identify seven most important changes which we then assess by more computationally expensive simulation techniques. Later, we confirm the results by an in vitro enzymatic assay. In this way, we identify five mutations that reduce the drug effectiveness. At the same time, some of them increase the native PLpro activity. These two facts suggest that the identified genetic variants should be taken into account for designing and testing new SARS-CoV-2 drugs.

**Funding:** The Sulkowska lab is supported by the National Science Center grant UMO-2020/01/0/NZ7/00244 and 2018/31/B/NZ1/04016. The Drag lab is supported by the National Science Center grant UMO-2020/01/0/NZ1/00063, and the "TEAM/2017-4/32" project, which is carried out within the TEAM program of the Foundation for Polish Science, co-financed by the European Union under the European Regional Development Fund. The funders had no role in study design, data collection and analysis, decision to publish, or preparation of the manuscript.

**Competing interests:** The authors have declared that no competing interests exist.

## Introduction

The outbreak of coronavirus disease 2019 (COVID-19), caused by the novel severe acute respiratory syndrome coronavirus 2 (SARS-CoV-2), has significantly affected almost every aspect of daily life, from the economy to the health sector crisis. COVID-19 patients can suffer from a wide range of symptoms, with the most commonly characterized by fever, fatigue and dry cough, but acute respiratory problems, multiple organ failure, and death are also possible [1]. As of July 2022, the World Health Organization (WHO) reported a cumulative total of over 550M confirmed cases with more than 6.3M deaths [2]. Currently, there is only one antiviral drug for COVID-19 approved by the U.S. Food and Drug Administration (FDA)—Veklury containing remdesivir as its active ingredient. The second FDA-authorized drug, Olumiant (baricitinib), does not affect the virus itself [3]. Therefore, there is a huge amount of research focused on finding new COVID-19 treatments [4, 5]. Additionally, there are three FDA-approved COVID-19 vaccines produced by Pfizer-BioNTech, Moderna and Janssen [6–8]. However, there are concerns whether the existing and the newly designed treatments are equally effective against different SARS-CoV-2 genetic variants.

SARS-CoV-2 is a positive-sense single-stranded RNA virus from the *Coronaviridae* family [9]. Generally, due to the lack of a proofreading mechanism, RNA viruses can mutate with an exceptionally high rate, which contributes to their enhanced virulence [10]. However, *Nidovirales*, an order of enveloped, positive-strand RNA viruses, which includes the *Coronaviridae* family, remains the exception. The viruses of this order possess a proofreading 3'-to-5' exoribonuclease (ExoN), which provides an improved replication accuracy and genome expansion [11–13]. SARS-CoV-2 variants are reported to emerge at a moderate rate [14, 15], some of which have been thoroughly investigated due to their potential association with reduced efficacy of treatments or enhanced disease severity [16]. One of the coronavirus strains that has temporarilly dominated the pandemic is the B1.351 lineage, which was first identified in South Africa [17]. The variant is reported to have $\sim$50% increased transmission rate [18] and its neutralization by post-vaccination antibodies and convalescent sera is even reduced by more than 10-fold [19, 20]. It was also indicated that some SARS-CoV-2 mutations may reduce its susceptibility to therapeutics targeting the viral RNA-dependent RNA polymerase, namely FDA-approved remdesivir [21, 22]. Therefore, mutation analysis is not only essential for tracking virus's evolvability but it may also be a crucial step toward the development of new vaccines and antiviral drugs.

The SARS-CoV-2 genome contains 12 functional open reading frames (ORFs) which encode 16 non-structural proteins (NSPs) and four structural proteins [23]. ORF1a and ORF1b are translated as viral polypeptides that have to be cleaved by main protease (Mpro) and papain-like protease (PLpro) to become functional peptides [24]. Mpro and PLpro are crucial for successful viral replication and therefore have become potential drug targets [25, 26]. The strategy of inhibiting the protease responsible for polypeptide cleavage has been successfully applied in the treatment of hepatitis C virus (HCV) [27] and human immunodeficiency virus (HIV) [28]. Beyond the polypeptide processing function, PLpro is also involved in reversing certain post-translational modifications of host proteins. It removes interferon-stimulated gene product (ISG15) but it can also cleave a bond between ubiquitin (Ub) and a host protein [24, 29]. ISGylation and ubiquitination mediate the innate antiviral defense, therefore PLpro actions lead to the suppression of the host immune response [29, 30]. These features of PLpro make the protease an ideal drug target for development of the COVID-19 treatment.

PLpro is encoded in the NSP3 region of ORF1a, which was demonstrated to be a site with the largest number of missense variants out of 16 ORF1ab proteins [14, 31]. Moreover, some substitutions were proved to reduce activity of SARS-CoV and SARS-CoV-2 PLpro with Ub

and ISG15 [29, 32–34]. Since mutations can also affect the drug binding affinity, it is important to conduct variant studies while designing therapeutics, especially for targets with high substitution rates, such as PLpro.

The main strategy for the design of new agents aiming to block SARS-CoV-2 PLpro activity is the development of noncovalent inhibitors. The current epidemiological situation and PLpro's significant druggability provided this enzyme with a great scientific interest. Thus, multiple PLpro inhibitors have been found [24, 35, 36] and new potential ones are being constantly proposed [26, 37]. Although the newly designed compounds exhibit some structural differences, they share multiple common characteristics and the majority of them may be divided into two main types in terms of the chemical structure of the core of the molecules. The first one includes derivatives of N-[1-(naphthalen-1-yl)ethyl]benzamide, with the most prominent representative—GRL-0617 [24]. The second group consists of derivatives of N-benzyl-1-[1-(naphthalen-1-yl)ethyl]piperidine-4-carboxamide (e.g. rac3j) [36]. In order to properly cover the chemical diversity of the PLpro inhibitors, we picked representative compounds, along with protein-ligand complexes, for both types, retrieved from the Protein Data Bank (PDB). Specifying, we chose GRL-0617, as the most important molecule from the first type of the noncovalent inhibitors, and N-[3-(acetylamino)benzyl]-1-[(R)-1-(1-naphthyl)ethyl]piperidine-4-carboxamide (S43 from PDB ID: 7e35), as the only representative of the second one that was also crystallized with SARS-CoV-2 PLpro.

In this work, we investigated the effects of selected PLpro mutations on binding affinity toward its known, representative inhibitors, namely GRL-0617 and S43. We based our study on over 2.5M PLpro sequences retrieved from the National Center for Biotechnology Information (NCBI) Virus Database. After an initial statistical analysis of the amino acid variants (single, double and triple) along the entire PLpro sequence, we focused on the mutations at the inhibitor binding site. We employed theoretical calculations to assess the impact of each substitution using molecular docking and binding energy calculations. As a result, we selected seven mutations that showed the most significant negative effects on the protein-ligand binding and assessed them by molecular dynamics (MD) simulations. We verified our predictions with an *in vitro* enzymatic assay.

## Results and discussion

SARS-CoV-2 is undergoing massive academic evaluations, that bring vast amount of data to analyze including its genetic sequence. Due to the pandemic character of the virus, the sequences are gathered from all-around the world and then deposited in the databases for analysis. We used NCBI Virus database and ORF1a amino acid sequences, that include papain-like protease (PLpro) domain, to evaluate its sequential variability in the human population. Our main focus is to assess if the current mutations of the PLpro are present in the inhibitor binding site and if so, what their impact is on the protein-ligand interactions.

Out of 2,610,999 PLpro sequences, we found mutations in 21% of them (540,129; see S1 File for detailed information about all of the variants). In most sequences only a single mutation is found in the sequence (86.5%), less frequently at two (12.8%), three (0.5%) and more (0.02%) positions.

### The variants are dispersed evenly in the PLpro sequence

Structure of papain-like protease is comprised of two main domains: ubiquitin-like (Ubl) domain and catalytic domain. The latter is further subdivided into the thumb and palm domains, which include the catalytic triad, and the fingers domain, which contains the zinc-finger motif (Fig 1). The inhibitor binding site is located between the thumb and the palm

domains. The most crucial part of this site, in terms of inhibitor interactions [38, 39], is formed by residues of the blocking loop (BL2) that acts as the lid for the site. After mapping the obtained sequence data on PLpro structure from PDB, we determined whether there are regions in the protein that are particularly susceptible to mutations. Due to the fact that the vast majority of PLpro amino acids can be mutated (98%), no particular region is more affected by modifications than others (Fig 1). We found only five amino acids that are not being mutated in any of the sequences (Y95, L118, Y137, H272, Y273). That includes H272—one of the catalytic residues. Possibly, other invariant amino acids are important for the proper functioning of the protein. Therefore, analysis of non-mutated amino acids of unknown function will help identify positions or regions of interest in drug design that are important to the virus. Interestingly, apart from H272 the rest of the known functionally important amino acids like zinc-binding cysteines (C189, C192, C224, C226) or protein-binding residues (D164, E167) are not invariant. However, their mutation rates are minuscule (see S1 File for more details).

## The most frequent PLpro mutations

The A145D mutation is the most prevalent variation in the PLpro sequence, present in about 36% of all the sequences that deviate from the reference. Most often it is the only variant in the sequence, but it can also occur together with more mutations. We observe a great diversity among the mutations with which A145D co-occurs—which shows that there is no amino acid whose mutation is dependent on the presence of the A145D variant and vice versa. A145 is located on the surface of the protein and is surrounded mostly by polar residues, including three lysines with no other contacts. Introduction of the aspartic acid may create new strong interactions with the positively charged residues, such as K91 or K92. The mutation to Asp is not an only option seen for this amino acid, we observed also six different variants in this position (V, S, T, G, P, N; Table 1). The mutations at this position first appeared in late 2020 (October) and quickly expanded over the following months to become the most frequent PLpro mutation (S1 Fig; this accords with earlier findings [40]). A145 mutation reached its peak of popularity in April and May 2021, when more than 60% of the mutated PLpro sequences carried it. However, in these months we see the highest percentage of mutated sequences—up to 80% of all the sequences that were deposited in the NCBI database were carrying at least one mutation. In our data set such situation appeared for four consecutive months (April-June 2021) and was never repeated (S1 Fig).

The second most frequently changed amino acid (P77) is present in 27% of the mutated sequences. Similarly to A145, it can occur as a single mutation, but also as one of many mutations. In addition, there are numerous variants associated with P77, indicating that this mutation is not co-dependent on other positions.

P223 is another amino acid that began to mutate more frequently in late 2020, which is consistent with the earlier research [40]. This amino acid is important for protein function by being in the zinc-binding loop (right next to cysteine directly interacting with the ion) and also as a part of the S1 ubiquitin binding site. Proline located at this position can be changed to one of five amino acids: leucine (the most common variant), serine, histidine, phenylalanine, or threonine. Any of these variants can make a great impact on the region since the observed amino acid change is significant.

## Mutations at inhibitor binding site

Amino acids that can undergo mutation are dispersed evenly throughout the protein sequence, including the binding site for the inhibitors. Based on all available PLpro-ligand crystal

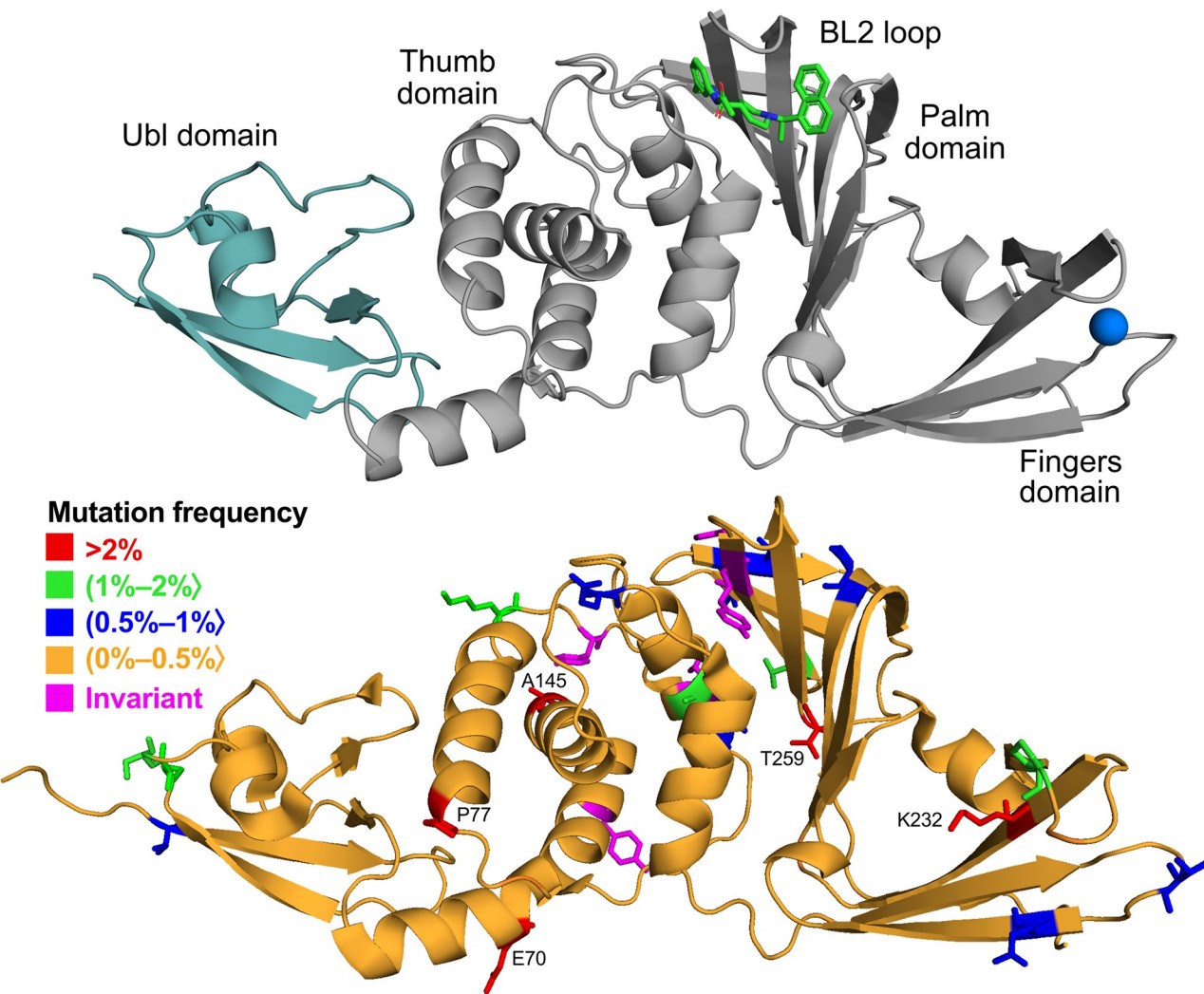

**Fig 1. Structure of PLpro and location of the mutations.** Upper panel: cartoon representation of the protein with inhibitor S43 shown in sticks and zinc ion as a sphere (based on PDB ID: 7e35). Teal region of the protein indicates N-terminal ubiquitin-like domain (Ubl). Lower panel: cartoon representation of the protein with residues colored according to their mutation frequency (percentage of sequences with a given mutation out of all mutated sequences).

structures in PDB, we identified the residues interacting with the inhibitors (both covalent and noncovalent; Table 1). Most importantly, there are many different variants of 22 amino acids present in the site (Fig 2). The affected residues spread out evenly across the site and can affect binding of the molecules in various ways. Even though in this work we focus on the noncovalent inhibitors, we utilized the information about the residues obtained from complexes with covalent ligands as well. It is because we believe that the current set of inhibitors represents just a beginning of the search for potent PLpro inhibitor and it is crucial to gather and present the entirety of the data.

**Covalent inhibitors.**   Based on the currently available crystal structures, some of the residues we found to be part of the inhibitor binding site are only interacting with covalent inhibitors (W106-Y112; PDB ID: 6wuu, 6wx4). Because the inhibitors form a bond with C111, all the surrounding residues can be important for the molecule's binding; yet, the mutation rate of these residues is exceedingly low. In particular, despite its low frequency, N109S variant

**Table 1. Selected amino acid variants of papain-like protease from SARS-CoV-2.** Frequent mutations are those with mutation rate >0.002. Residues from inhibitor binding site were selected based on residue-ligand distance (no more than 6 Å) calculated for all available PLpro-ligand crystal structures. Amino acids in Variants column are sorted from most to least frequent. All found variants are available in Supplementary Material.

| Amino acid | Position (PLpro) | Position (NSP3) | Position (ORF1a) | Variants | Number of mutated sequences | Mutation Rate |
|---|---|---|---|---|---|---|
| **Inhibitor binding** | | | | | | |
| A | 107 | 852 | 1670 | V, T, S, Q, E | 1624 | 0.00062 |
| D | 108 | 853 | 1671 | A, G, Y, N, E, V, K | 281 | 0.00010 |
| N | 109 | 854 | 1672 | D, S, K | 76 | 0.00003 |
| Y | 112 | 857 | 1675 | H,C,F,N | 48 | 0.00002 |
| K | 157 | 902 | 1720 | N, R, T, G, M | 1002 | 0.00038 |
| E | 161 | 906 | 1724 | D, G, A, K, V | 4819 | 0.00185 |
| L | 162 | 907 | 1725 | F, S, V, I | 48 | 0.00002 |
| G | 163 | 908 | 1726 | C, S, A, D | 311 | 0.00012 |
| D | 164 | 909 | 1727 | G, E, S, N | 17 | 0.00000 |
| V | 165 | 910 | 1728 | I, A, F, D, L | 424 | 0.00016 |
| R | 166 | 911 | 1729 | K | 11 | 0.00000 |
| E | 167 | 912 | 1730 | D, V, A, G, K | 16 | 0.00000 |
| M | 208 | 953 | 1771 | T, I, V, L, K, R, A, G | 1436 | 0.00055 |
| A | 246 | 991 | 1809 | T, V, S, E, G | 1130 | 0.00043 |
| P | 247 | 992 | 1810 | L, S, Q, T | 2012 | 0.00077 |
| P | 248 | 993 | 1811 | L, S, F, H | 144 | 0.00006 |
| A | 249 | 994 | 1812 | V, D, T, S, G | 1187 | 0.00045 |
| Q | 250 | 995 | 1813 | H, L, R, P, K, E | 1709 | 0.00065 |
| Y | 264 | 1009 | 1827 | H | 4 | 0.00000 |
| G | 266 | 1011 | 1829 | D | 9 | 0.00000 |
| N | 267 | 1012 | 1830 | D, S, T, K, I, Y | 260 | 0.00010 |
| Y | 268 | 1013 | 1831 | H, C, F, D, N | 123 | 0.00005 |
| Q | 269 | 1014 | 1832 | K, L, R, H | 147 | 0.00006 |
| C | 270 | 1015 | 1833 | Y, R, F, G, S, W | 749 | 0.00029 |
| P | 299 | 1044 | 1862 | S, L, H, T, R | 3657 | 0.00140 |
| T | 301 | 1046 | 1864 | A, M, S, R | 48 | 0.00002 |
| D | 302 | 1047 | 1865 | N, E, M, Y | 99 | 0.00004 |
| **The most frequent mutations** | | | | | | |
| A | 145 | 890 | 1708 | D, V, S, T, G, P, N | 193489 | 0.07411 |
| P | 77 | 822 | 1640 | L, S, H, F, T, R, I | 147359 | 0.05644 |
| E | 70 | 815 | 1633 | D, G, Y, A, K, V | 32190 | 0.01233 |
| K | 232 | 977 | 1795 | Q, T, N, R, E, I | 26076 | 0.00999 |
| T | 259 | 1004 | 1822 | I, A, N, S, V | 12808 | 0.00491 |
| T | 277 | 1022 | 1840 | I, N, A, F, S | 10561 | 0.00404 |
| A | 116 | 861 | 1679 | S, V, T, E, G | 9614 | 0.00368 |
| S | 24 | 769 | 1587 | S, P, A, T, N, Y | 7360 | 0.00282 |
| M | 23 | 768 | 1586 | I, V, L, T, K | 6666 | 0.00255 |
| K | 92 | 837 | 1655 | N, R, E, Q, T, M, G | 6510 | 0.00249 |
| P | 223 | 968 | 1786 | L, S, H, F, T | 6013 | 0.00230 |
| T | 191 | 936 | 1754 | I, N, A, S, P | 5963 | 0.00228 |

could be important, since the mutation to serine introduces a smaller residue, which could diminish the interactions between the molecules.

**Noncovalent inhibitors.** Our main focus is to identify residues that strongly reduce the inhibitory effect of noncovalent molecules. In order to identify such residues, we first analyzed

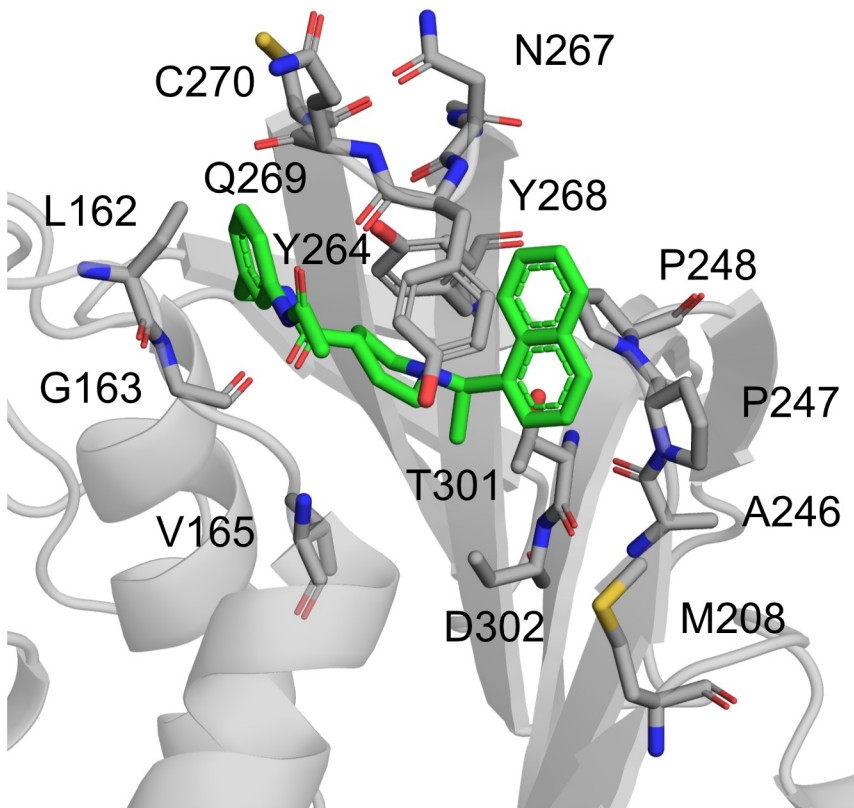

**Fig 2. Structure of PLpro and variants within noncovalent inhibitor binding site.** The rates of mutations of these residues are presented in Table 1, along with the residues they are replaced to. It is worth noting that the residues forming the BL2 also can be mutated (aa 267–270). Based on PLpro-S43 complex from PDB ID: 7e35.

PLpro-ligand interactions in wild type (WT) complexes with two chosen representatives: S43 (PDB ID: 7d7t) and GRL-0617 (PDB ID: 7jrn). In the crystal structure of the protein-S43 complex, the ligand interacts with the protein via single hydrogen bond with main chain of Y268 and non-polar interactions with P247, P248 and Y268. In the case of the crystal containing GRL-0617, there is also a single hydrogen bond (but with D164) and the hydrophobic interactions involve P248 and Y268. Hydrogen bonding with side chain of D164 is not possible for S43 because the ligand's polar groups are not properly aligned for such an interaction. Both ligands contain naphthalene, which is involved in the $\pi - \pi$ interactions with the protein where it forms T-shaped stacking with Y268.

Next, in order to gather more in-depth information about protein-ligand interactions that goes beyond static crystal structures, we performed multiple 500 ns long Molecular Dynamics (MD) simulations of both complexes. In the case of GRL-0617, the hydrogen bond present in the crystal structure is poorly maintained in the WT simulation of the complex (6% of the simulation time). The main hydrogen bond found in the trajectories is with the main chain of Q269 (over 85% of the time). Bonds between the ligand and side chain of Q269, Y264, and Y268 are significant as well (over 40%, 25%, and 10%, respectively). Overall, the average number of hydrogen bonds is 1.7. In over 80% of the simulation time naphthyl group has $\pi$-alkyl interaction with P248 and in over 20% $\pi - \pi$ with Y268.

GRL-0617 is maintained in the binding site by both polar and hydrophobic interactions, whereas most of the S43 and protein binding interactions are hydrophobic. However, the

hydrogen bond present in the crystal structure is further maintained in the simulation of the complex. Bonds with other amino acids are scarcely present—about 10% of the simulation time with side chain of Q269 and up to 10% with main chain of L163. The hydrophobic interaction network is more extensive. In over 80% of the simulation time naphthyl group has nonpolar interaction with P248, over 25% with Y268 and over 20% with P247. Moreover, the piperidine ring of S43 interacts for 50% of the time with Y268. In the Wild-Type trajectories additional interaction with Y264 emerged that was not present in the crystal structure (piperidine-Y264 for about 60% of the simulation time).

This analysis shows the importance of five amino acids in particular: P247, P248, Y264, Y268, and Q269. They have high impact on the ligand binding energy (S5 Fig). Surprisingly, almost all of them were found to be mutated in the data set we analyzed. P247 can be changed to one of four different amino acids and importantly, the change is substantial—from small non-polar proline to polar serine or bigger glutamine. Given that this amino acid interacts with the aromatic rings of ligands, such a change could have negative effects on their binding. However, the most common mutation is P247L, which we predict would not have a severe impact on the protein. Similar findings are applicable for P248, that is most frequently mutated to Leu and Ser. In the case of Y264, we only detected four sequences with its variant and it always co-occurs with a mutation in E263 (E263D-Y264H). The remaining two amino acids (Y268 and Q269) are already established as the most important amino acids, capable of interacting with a variety of inhibitors [41, 42]. It was shown that both Y268G and Y268T negatively affect the inhibition of GRL-0617 [29]. However, we do not observe these specific amino acids, but rather changes to His and Cys at position 268 and to Lys, Leu and Arg at position 269. Interestingly, although mutations of Y268 or Q269 are not common, we observe double and triple variants involving these residues (e.g. T265A-Y268C).

## Selecting variants potentially important for inhibitor binding

First, in order to choose variants of amino acids from the PLpro binding site worth further investigation, we roughly estimated the impact of the mutations on the inhibitor binding. For this purpose, we utilized molecular docking and simplified binding energy calculations in Discovery Studio. We took into account most of the mutations that had been encountered up-to-date for the residues in the proximity of the inhibitor binding site. We tested their effects on two PLpro-inhibitor complexes, containing GRL-0617 and S43 (PDB IDs: 7jrn and 7d7t, respectively). We utilized three main methods. The first one is the estimation of the mutation energy, namely the difference between the mutant and the wild type (WT) binding energies. The latter two included redocking to the mutated PLpro, prepared in two ways. Additionally, to ascertain the results, we employed additional scoring functions, MMGBSA binding energy calculations, and one more PLpro-inhibitor complex, PDB ID: 7jn2 (see S2 Fig for more information).

Results of both docking and energy calculations highlight the mutations that may negatively affect potential drugs' binding. The three probably most significant ones are P248S, and the double mutations E263D-Y264H and T265A-Y268C (Table 2). The latter variant most strongly affects the inhibitors' binding. The results from the additional calculations support these findings. Moreover, they indicate that mutations of P247 and the double variant E161D-P247S may be of importance, although it seems less probable than the aforementioned ones.

One has to bear in mind that techniques such as docking are inaccurate to semi-accurate. Thus, in some cases they may also produce results that will turn out to be false positives. In order to reduce this risk, we utilized two docking protocols and simplified mutation energy

**Table 2. Results of the estimated impact of the mutations at the PLpro binding site on the inhibitor binding.** For each PLpro-inhibitor complex, the first column shows mutation energy, which is the difference of the binding energies between the respective mutant and the wild type (WT) protein, estimated using Calculate Binding Energies protocol in Discovery Studio. The next two columns include negatives of the CDOCKER interaction energies from molecular docking conducted to the PLpro mutants prepared with two methods in Discovery Studio. The first one utilized the Build Mutant protocol. The second one consisted of the mutation of specific residues with a subsequent side chain conformation refinement.

| | PLpro—GRL-0617 | | | PLpro—S43 | | |
|---|---|---|---|---|---|---|
| | | −CDOCKER interaction energy (kcal/mol) | | | −CDOCKER interaction energy (kcal/mol) | |
| Mutation | Mutation energy (kcal/mol) | Method 1 | Method 2 | Mutation energy (kcal/mol) | Method 1 | Method 2 |
| WT | - | 46.0 | 46.0 | - | 53.2 | 53.2 |
| P129S-V165A | 0.0 | 44.3 | 46.6 | 0.0 | 53.4 | 53.2 |
| A145D-G163C | -0.2 | **39.0** | 47.1 | -0.1 | 52.8 | 53.5 |
| A145D-A246T | -0.1 | 44.2 | 45.5 | -0.1 | 55.7 | 55.2 |
| E161A-T291I | 0.1 | **39.3** | 45.0 | 0.1 | 54.4 | 55.3 |
| E161D | 0.0 | 44.1 | 47.1 | 0.0 | 53.9 | 56.4 |
| E161D-P247S | **0.4** | 45.6 | 46.5 | 0.3 | 58.0 | 56.3 |
| E161G | 0.1 | 44.1 | 45.8 | 0.2 | 54.0 | 55.3 |
| L162S | 0.3 | **39.0** | 46.2 | **1.3** | 54.0 | **50.8** |
| G163S | -0.1 | 44.2 | 46.2 | 0.0 | 53.3 | 52.1 |
| V165I | 0.0 | 45.1 | 45.7 | 0.0 | 55.5 | 54.4 |
| A246V | -0.1 | 45.6 | 45.7 | -0.1 | 53.8 | **50.1** |
| P247L | -0.2 | 45.2 | 46.4 | -0.3 | 56.0 | **47.0** |
| P247Q | 0.3 | 46.4 | 46.2 | 0.1 | 56.5 | 52.9 |
| P247S | **0.4** | 46.3 | 44.3 | 0.3 | 56.3 | 51.3 |
| P247T | 0.2 | 45.8 | 45.5 | 0.1 | 55.6 | 53.7 |
| P248S | **0.6** | **41.7** | **43.6** | **0.4** | **50.9** | 51.4 |
| E263D-Y264H | **0.7** | **40.2** | **39.4** | **0.5** | 54.0 | **51.0** |
| T265A-Y268C | **1.4** | **38.4** | **40.9** | **2.5** | **50.5** | **46.1** |
| N267D | -0.1 | 45.8 | 45.7 | -0.2 | 54.4 | 52.6 |
| N267S | 0.0 | 44.2 | 47.5 | 0.0 | 55.4 | 54.3 |
| Y268H | 0.3 | 46.5 | 46.7 | **0.7** | **50.5** | 51.9 |
| Q269K-T277I | 0.2 | 44.6 | 48.0 | 0.0 | 53.2 | 52.8 |
| Q269R | -0.4 | 45.0 | 46.2 | 0.1 | 53.5 | 53.4 |
| T301A | 0.2 | 45.2 | 44.0 | 0.2 | 52.6 | 52.8 |
| T301S | 0.1 | **39.3** | 45.8 | 0.1 | 54.8 | 52.9 |

calculations. Merging results from multiple, even semi-accurate techniques, allows to obtain a consensus view that is less prone to feature a random, false outcome. Nevertheless, despite the possible problems with false positive results, such methods are valuable because of their ability to initially screen out mutations unlikely to significantly affect potential drugs binding. For example, based on PLpro crystal structures, Q269 is one of the key residues when it comes to binding site steric properties and creating interactions with the inhibitor. Therefore, the mutation of this amino acid should intuitively have a significant impact on ligand binding. However, both MD and the faster methods show that Q269R does not substantially affect the inhibitors' affinity to PLpro. Thus, techniques such as docking may effectively act as a preliminary sieve and allow to select only most probable candidates for the more time-costly evaluation.

## Impact of selected variants on inhibitor binding

Taking all of the above analysis into consideration, we chose five positions in the PLpro, which mutations have the biggest potential to negatively impact inhibitory effects and binding of the

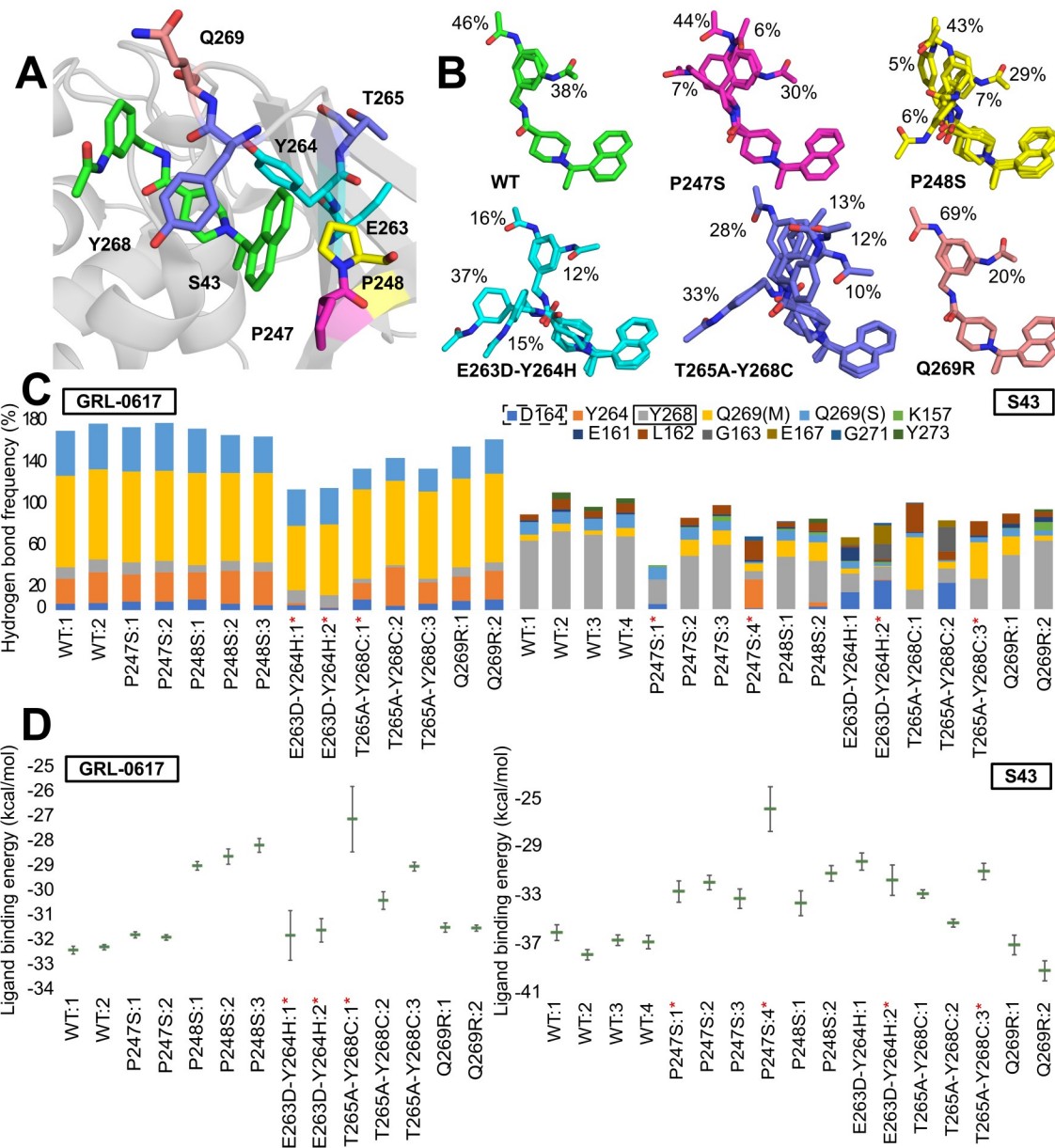

**Fig 3. Selected variants found in PLpro.** (A) Location of amino acids subject to mutation relative to the ligand (S43) binding site (based on PDB ID: 7d7t). (B) Representative S43 conformations bound to PLpro. Calculations are based on frames sampled every 100 ps from joint Molecular Dynamics trajectories. Conformations found in less than 5% of the frames are not shown for clarity. (C) Frequency of protein-ligand hydrogen bonds in different trajectories. Left: GRL-0617, right: S43. Calculations are based on frames sampled every 100 ps from Molecular Dynamics trajectories. Dashed rectangle points to the amino acid interacting via hydrogen bond with GRL-0617 in the crystal structure (PDB ID: 7jrn), the one with solid line with S43 (PDB ID: 7d7t). In the case of Q269 the ligand can bind to either its main chain (M) or side chain (S). The red asterisk indicates the trajectories in which the ligand dissociated from the binding site. (D) Ligands MMGBSA binding energy (kcal/mol) to PLpro. Left: GRL-0617, right: S43. Calculations are based on frames sampled every 100 ps from Molecular Dynamics trajectories—the averages and standard errors are shown. The trajectories in which the ligand dissociated from the binding site are marked with red asterisk.

ligands (P247, P248, Y264, Y268, Q269; Fig 3A). For each selected variant, we performed all-atom molecular dynamics simulations (at least two trajectories for 500 ns) and calculated ligand binding energy using MMGBSA for two inhibitors: GRL-0617 and S43. Overall, for ten different variants we performed over 30 trajectories. Along with the variants predicted to be

impactful, we also analyzed three additional variants (G163S, A246V and N267D) that served as a quality control. We measured the stability of the whole protein and the binding site using RMSD and it shows similar behavior of the mutants in regard to WT (S3 and S4 Figs). The differences we observed are discussed in detail below.

**P247S.** Out of the possible variants in position 247 the one with the serine replacing proline has the biggest impact on the inhibitor binding. For the S43 ligand, the binding energy ranges between -25.8 ± 1.8 kcal/mol to -33.1 ± 0.8 kcal/mol, which is significantly higher than the energy observed in WT simulations. Conformation of the ligand in its binding site is mostly resembling that observed in wild type conditions (Fig 3B). However, there are two new representative conformations that are showing changes that appear due to the mutation. For example, we observed two new bonds formed between the ligand and amino acids D164, Y264 (Fig 3C). The main hydrogen bond with Y268 seen in the WT simulation is greatly reduced, especially in the trajectories with ligand leaving the binding site. Overall, the average number of hydrogen bonds in the trajectories is slightly smaller than in WT as it ranges from 0.5 to 1. Due to the lack of proline in 247 position, nonpolar interactions between ligand and Y264 and Y268 are affected. Specifically, very weak interactions between naphthalene and piperidine rings and Y268 seem to be connected to ligand exiting the site, as these interactions are present in P247S trajectories where the ligand is bound the entire time.

On the other hand, GRL-0617 is stably bound to the binding site for the entire trajectories. Ligand's binding energy based on MMGBSA calculation (-31.7 ± 0.1 and -31.8 ± 0.1 kcal/mol; Fig 3D) is not different from the one obtained for WT trajectory. This shows that the mutation does not have an impact on the protein-ligand interactions. Specifically, GRL-0617 does not change its conformation during any of the trajectories. Both polar and nonpolar interactions between protein and the ligand are stable. Moreover, due to the fact, that the important nonpolar interactions with naphthyl group are mainly realized by P248, mutation of P247 does not affect the non-polar interactions between protein and the ligand.

**P248S.** Regardless of the type of ligand bound to the protein containing the P248S variant, the ligand did not dissociate in either of them. However, in both cases, the energy of the ligand bound to the mutated protein is less favorable than in WT, which shows that this variant might affect the protein's binding capabilities in a universal way, regardless of the ligand.

In trajectories with bound GRL-0617, the ligand binding energy ranges from -28.1 ± 0.3 to -28.9 ± 0.2 kcal/mol (Fig 3D), which is higher than the energy observed for the WT protein. Throughout the trajectory, the ligand is stable and maintains its native conformation. Similarly, the polar interactions between the protein and the ligand resemble those found in the WT protein. The only difference is the slightly lower frequency of interactions with the Q269 side chain. However, the main effect of the lack of proline at position 248 is seen when analyzing non-polar interactions, due to the lost P248 interactions with the naphthalene rings. Their absence is offset by the more frequent naphthalene-Y268 interaction.

In the case of the protein-S43 complex, the ligand binding energy is also higher than in the WT protein (-33.5 ± 1.0 kcal/mol and -31.1 ± 0.6 kcal/mol), indicating a negative effect of the P248S variant on S43 function. There are only minor differences in polar protein-ligand interactions between the mutant and WT proteins. Lower hydrogen bonding rates with Y268, but slightly higher with the main chain Q269. Overall, the average protein-ligand hydrogen bond number (0.9) remains similar to that of WT (1.1). There is an apparent weakening of the interactions between naphthalene and P247 and Y268. However, Y268 remains in contact with the piperidine ring. Most of the time, the ligand adopts the same conformations as in the not mutated trajectories.

Overall, the P248S variant weakens the non-polar interactions between protein and ligand. However, even though the majority of protein-ligand interactions were non-polar, the ligands

did not dissociate from the binding site. The importance of the residue for the ligand binding is high—the variant is not able to match the energy composition of the native proline. Also, it affects the P247 contribution (S5 Fig).

**E263D-Y264H variant.** To assess the impact the double variant E263D-Y264H has on the protein-ligand interactions, we performed similar MD simulations as we did with WT protein. Although the detailed results differ between the two ligands studied, the ultimate outcome is the same: impaired protein binding that leads to dissociation from the binding site.

For the S43 ligand, the binding energy is higher than in WT protein, showing that S43 is affected by E263D-Y264H variant (-30.1 ± 0.7 kcal/mol and -31.6 ± 1.2 kcal/mol). It is manifested by weaker stabilization of the molecule in its binding site. Ligand S43 changes conformation, which is shown by substantially higher Root Mean Square Deviation (RMSD) than in WT trajectories (4 Å and 5 Å for trajectories 1 and 2, respectively). Despite such high fluctuation values, S43 dissociates from the binding site only from trajectory 2 (after 415 ns). In terms of protein-ligand interactions, we observed that new bonds are formed between main chain of D164, E167 and the ligand (Fig 3C). The main hydrogen bond with Y268 seen in the WT simulation is greatly reduced, to only 12% and 18% for the two trajectories. Overall, the average number of hydrogen bonds in the trajectories is slightly smaller than in WT—0.7 and 0.9. Moreover, there are also slightly weaker interactions between naphthalene and two prolines (P247, P248) than in WT. New interaction between benzene ring and K157, not present in WT trajectories, is a result of ligand's change in conformation in the binding site (Fig 3B). Also, the mutation of Y264 impacted its $\pi$-alkyl interaction with the piperidine ring of S43 ligand.

The ligand is dissociating from the binding site in both trajectories for the E263D-Y264H mutant with bound GRL-0617 (after 194 ns and 370 ns). The energy of the ligand binding in this variant is similar to the energy from WT trajectory (-31.7 ± 1.0 kcal/mol and -31.5 ± 0.5 kcal/mol). Even though the ligand exits its binding site in both trajectories, $C_\alpha$ RMSD of the protein remains stable and fluctuating around 2 (S3 Fig). The mutated residues are not inducing any substantial change to the protein's backbone. Also, ligand is not changing its conformation (S6 Fig) and only slightly bigger fluctuation of the 4-methylbenzenamine part of the ligand can be observed. Due to the mutation of Y264 to histidine the rate of the hydrogen bonds with this residue dropped from over 25% to 2%. Similarly, hydrogen bond between main chain of Q269 and the ligand is lower by over 20%. Overall, the number of protein-ligand hydrogen bonds is smaller in this variant (on average 1.2) than in WT (1.7). In terms of protein-ligand stacking interactions, there are only small differences. This variant attributes to more frequent stacking interaction between Y268 and inhibitor's naphthyl group than in WT. The reason behind ligand's dissociation from the protein most probably lies in a lower number of hydrogen bonds, mainly with mutated 264 amino acid.

**T265A-Y268C.** Out of two substitutions in this double variant, the mutation of Y268 has bigger impact on the inhibitor binding, due to its non-polar interactions with the ligand. We observe dissociation in one of three trajectories, regardless of which ligand is bound to the protein.

For the S43 ligand, the binding energy ranges between -30.9 ± 0.7 kcal/mol to -35.1 ± 0.3 kcal/mol, which is slightly higher than the energy observed in WT simulations. Conformation of the ligand significantly differs from that observed in WT trajectories (Fig 3B). The two main WT conformations are present in only 38% of the time. Moreover, the most frequent conformation is entirely different and found in 33% of the time. For this mutation, we observe higher frequency of interaction with main chain of Q269 (Fig 3C). The main hydrogen bond of WT trajectories that involve mutated Y268 is greatly reduced. Overall, the average number of hydrogen bonds in the trajectories is similar to WT (around 1). Due to the lack of the tyrosine,

stacking interactions between ligand and the protein are affected. Also, we observe less frequent interactions between ligand's naphtyl group and P247, P248 and between Y264 and piperidine ring of S43.

Similar circumstances are observed in the trajectories of protein-GRL-0617 complex. Ligand's binding energy based on MMGBSA calculation (-27.0 ± 1.3 and -30.3 ± 0.3 kcal/mol; Fig 3D) is higher than that of the WT trajectory. Even though GRL-0617 does not change its conformation during any of the trajectories, the hydrogen bonds between protein and the ligand differ from those found in not mutated complex. The interactions between Y264 and side chain of Q269 are diminished. In the case of the non-polar ones, the naphtyl group interacts with P248 slightly less frequently.

**Q269R.** Binding energies of S43 ligand based on MMGBSA calculation (-36.9 ± 0.8 and -39.1 ± 0.8 kcal/mol) are slightly lower than that of the WT trajectory. This shows that certain favorable interactions are formed as a result of the mutation. Interestingly, neither protein-ligand hydrogen bonds nor stacking interactions significantly differ from WT protein. However, due to the mutation to arginine, a new type of interaction emerges—cation-$\pi$, which is frequently utilized by the ligand (up to 48% of the trajectory).

Even though Q269 is an important residue in protein-ligand interaction its mutation to arginine does not affect strength of ligand's binding. We performed two 500ns simulations of Q269R variant with bound GRL-0617 ligand. Ligand's binding energy based on MMGBSA calculation (-31.4 ± 0.2 and -31.4 ± 0.1 kcal/mol) is not different from the one obtained for WT trajectory. Moreover, since the ligand does not change its conformation during the trajectory, the hydrogen bond network in this variant is similar to the one present in WT protein. The effect of the mutation is not visible here as the ligand interacts with arginine and is able to maintain similar average number of hydrogen bonds during the trajectory as in WT. Stacking interactions are as well not affected by the presence of the variant. Even though, interactions between P248 and naphthalene ring are slightly less frequent, the mutation to positively charged arginine introduces the possibility of cation-$\pi$ interaction. Since the ligand has 4-methylbenzenamine moiety in vicinity of the arginine, the interaction between these groups is formed and is fairly frequent (almost 45% of the trajectory).

## Impact of selected variants on enzyme activity

Using theoretical approach we showed several variants that can impact binding of the inhibitors. Next, we focused on their influence on protein's activity and inhibitor potency using experimental methods.

**Activity of recombinant PLpro mutants.** To assess the influence of mutations on the activity of recombinant SARS-CoV-2 PLpro mutants, we performed an enzymatic assay using synthetic, fluorogenic substrate Ac-LRGG-ACC, with structure based on a C-terminal epitope of ubiquitin and ISG15 protein—physiological substrates of PLpro [43]. Activity was determined by measuring the increase of fluorescence over time (RFU/s) and normalized to the activity of the WT presented as 100% (see Fig 4). We observed that P247 mutants generally possessed higher activity in comparison to WT protease. P247T mutant displayed the highest activity toward tetrapeptide substrate with 147.7% of WT's activity. For P247S and P247L we observed respectively 121.8% and 126.0% of activity. However, for other mutants, the enzyme activity diminished. Only the E263D single mutant preserved 50.2% of activity while for P248S and Y264H we observed no substrate hydrolysis. The experiment enabled selection of active variants for further evaluation of the influence of mutations on GRL-0617 and S43 noncovalent inhibitors binding.

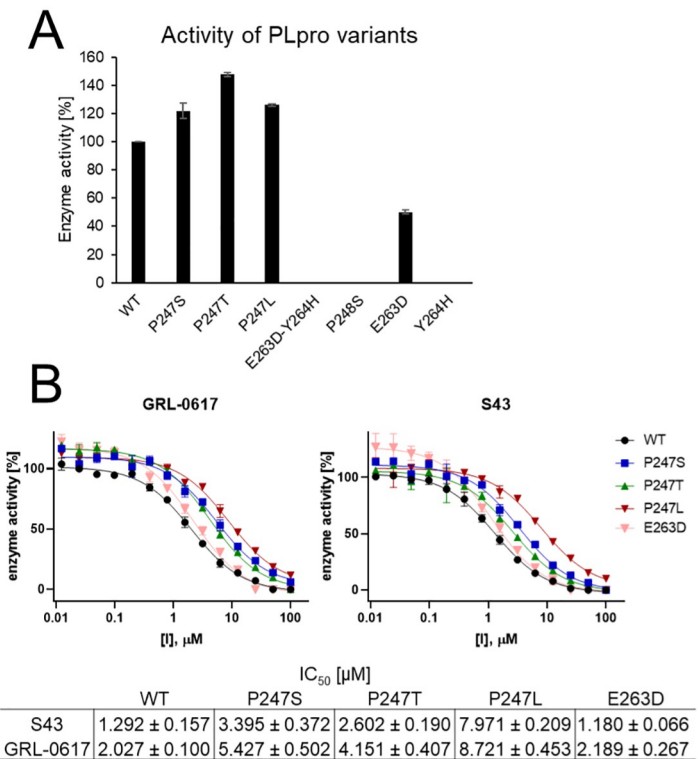

**Fig 4. Characterization of activity and inhibitor binding properties of recombinant SARS-CoV-2 PLpro variants.**
Assay conditions: [E]=100 nM, [S]=10 $\mu$M. (A) Determination of activity of the PLpro mutants toward tetrapeptide
fluorogenic substrate. Experiment was repeated at least two times. (B) IC50 determination for variants that possessed
enzymatic activity of GRL-0617 and S43 inhibitors. Enzyme was incubated in assay buffer for 10' followed by enzyme
incubation with inhibitor for 30' prior to the measurement. Measurements were performed three times.

**PLpro mutations influence the efficacy of inhibitors.**   *In silico* molecular dynamics simu-
lations revealed, that noncovalent binding of GRL-0617 and S43 inhibitors can be influenced
by particular mutations within the PLpro. We selected mutations at five sites, that should
mostly influence the potency of S43 and GRL-0617 noncovalent inhibitors, including P247,
P248, Y264, Y268, and Q269. To assess this influence *in vitro*, we expressed recombinant vari-
ants of the SARS-CoV-2 PLpro and performed an enzymatic assay to determine IC50 values of
the inhibitors against the active protease variants.

P247 mutations did not cause a loss of activity and molecular dynamics simulations for
P247S mutant predicted, that mutations at this site may influence binding of S43 but not GRL-
0617 ligand. However, during *in vitro* assays we observed that IC50 of both inhibitors for all of
the P247 mutants is significantly higher than for the WT. The biggest change was observed for
the P247L mutant, for which IC50 of S43 was 6 times higher (7.971 $\mu$M) and IC50 of GRL-
0617 was 4.5 times higher (8.721 $\mu$M) in comparison to the WT (respectively 1.292 $\mu$M and
2.027 $\mu$M). For serine and threonine mutants, the change of IC50 was smaller (2–3 fold
change) than for the leucine mutant.

MD simulations showed that the inhibitors dissociate from the E263D-Y264H mutant. We
were not able to validate this because we could not express the active double mutant variant of
the protein. All of the protein aggregated in inclusion bodies during expression procedure.
However, we successfully obtained active E263D mutant, which suggests the importance of

Y264 for protein functionality. Interestingly, *in vitro* assays using recombinant enzyme resulted with IC50 values almost the same as for the WT for both of the inhibitors.

## Conclusions

Millions of individuals throughout the world are affected by the COVID-19 pandemic. More mutations emerge in the SARS-CoV-2 genome as more people become infected. Even though these alterations only modify single residues in coronavirus proteins, such modifications can affect the virus's lethality, transmission, and susceptibility to vaccinations and drugs. Therefore, it is crucial to keep track of new mutations, because only then we will be able to quickly identify potentially dangerous mutations.

In this work, we identified new variants of amino acids in papain-like protease (PLpro)—many of them were not investigated before. We analyzed almost 70,000 PLpro sequences and found mutations in 5% of them. The sequences can differ on either one, two or three positions from the reference Wuhan PLpro from December 2019.

The sudden appearance and rapid spread of the mutation of alanine 145 (mostly A145D) suggests that it may become the first mutation in this protein to become conserved. Our analysis indicates that it appeared in October 2020, quickly dispersed and became the most frequent mutation in PLpro (present in 37% of the mutated sequences). Its popularity may be connected to the favorable interactions that it can build with surrounding lysines (K91, K92). Even though A145 seems not to be connected with substrate binding, further study is necessary to fully assess its function and potential.

Our main focus was to verify whether the identified PLpro variants have any impact on binding of the inhibitors and protein's overall activity. We used two representative molecules—GRL-0617 and S43, and estimated inhibitors' binding energy with two general methods: molecular docking (based on scoring functions and MMGBSA) and molecular dynamics (based on MMGBSA). Based on these results, we show that mutation P247S, and two double mutations: E263D-Y264H and T265A-Y268C, are the ones that should be monitored in the population. Based on the results of the *in vitro* experiments measuring inhibitors' potency and protein's activity, we see that some of the mutations can alter both. In particular, our results show the importance of position 247 in terms of protein functionality. Not only the mutation can lower the efficacy of the inhibitor, it also can increase enzymatic activity. Overall, we recommend including the variants in any computer-aided drug design (CADD) methods that target this protease.

The faster, more approximate methods, that we utilized before MD, proved to be a quick and efficient way to estimate which mutations are most likely to have a substantial impact on the inhibitor binding. Considering the fast rate of sequencing of SARS-CoV-2, it is crucial to test and develop protocols for rapid screening of potential variants. As the example of the mutation of A145 shows, if the variant poses any advantage to the protein and the virus, it can spread very effectively.

## Methods

### Finding amino acid variants in PLpro sequences

Total of 2,669,892 ORF1a sequences from human hosts were downloaded on June 29th 2022 from NCBI Virus database. As a reference we used ORF1a polyprotein from Wuhan, China (YP_009725295). Papain-like protease domain is located in this polyprotein between 1,564 and 1,868 amino acid and this sequence was later used as a reference PLpro sequence. Searching for amino acid variants in the sequences was performed using an in-house python script that first aligned each ORF1a sequence with the PLpro reference. We obtained results for

97.8% of the downloaded sequences (2,610,999). In the analysis, the positions with ambiguous symbols (X, B, J and Z) were omitted.

## Docking and mutation energy estimation

In order to estimate the effects of certain PLpro variants on the inhibitors' binding affinity, we employed three different methods using BIOVIA Discovery Studio v20.1.0.19295. Firstly, we checked the effects of mutations by conducting molecular docking. In order to generate the PLpro variants, we took two different approaches.

In the first method, we used the Build Mutant protocol which mutates selected residues and optimizes neighbouring amino acids within the distance cutoff. Considering the fact that selected mutations were between similar sized residues, we did not include any surrounding atoms for the optimization by setting the Cut Radius parameter to 0 Å. We created one model per each variant. Other parameters were set as default. Generated variants were then prepared by utilizing the Prepare protein protocol with CHARMm force field in the pH set to 7.4. We used a spherical grid with the radius of 15 Å created around the original ligand.

In the second method of creating mutants, we prepared the original protein and created a grid box using the same protocols as in the first method. Afterwards, we used the Build and edit protein tool to mutate particular amino acids and then used the Side-chain refinement protocol to find the best conformation of the mutated residues.

Before docking, we prepared the compounds by the Prepare ligands protocol and created possible ionization states for pH 7.5 ± 1 without generating tautomers or isomers. Then, we employed the CDOCKER protocol and retrieved only the best pose for each ligand by setting the option top hits to 1. We assessed the obtained poses by the Score ligand poses protocol and utilized the Calculate binding energies protocol with generalized Born implicit solvent model (MMGBSA).

Another approach taken for the analysis of PLpro variants was to calculate the difference in the inhibitor binding free energy between the variants and the wild type using the Calculate Mutation Energies (Binding) protocol. To conduct the estimations, we used complexes obtained from docking selected compounds to the wild type PLpro. The protocol was employed in the pH-independent mode, other parameters were set as default.

To conduct the calculations, we built PLpro variants based on crystal structures with PDB IDs: 7jrn (chain A), 7d7t (chain A), and 7jn2. We had validated CDOCKER's ability to correctly predict inhibitor's binding pose to 7jrn and 7jn2 in our previous work [26]. For this purpose, we had calculated the redocking RMSD values of ligand's heavy atoms (1.6 Å for 7jn2 and 2.7 Å for 7jrn). Chain A of 7jrn was chosen over the other based on the lower RMSD value (2.7 Å vs 2.8 Å for 7jrn chains A and J, respectively). In this study, we conducted analogical validation for 7d7t, with RMSD of 2.2 Å for chain A and 3.6 Å for B.

We conducted the analysis for 17 single mutations: E161D, E161G, L162S, G163S, V165I, A246V, P247L, P247Q, P247S, P247T, P248S, N267D, N267S, Y268H, Q269R, T301A, and T301S, and eight double mutations: P129S-V165A, A145D-G163C, A145D-A246T, E161A-T291I, E161D-P247S, E263D-Y264H, T265A-Y268C, and Q269K-T277I. The binding affinity estimations were made for four compounds with known, high in vitro activity against PLpro (GRL0617 [44], compound 6 [24], rac3j_R, rac3k_R, [36]) and noncovalent inhibitors bound to the crystal structures with PDB IDs: 7jn2 (PLP_Snyder441) and 7d7t (S43). For assessing the 7jrn-derived docking poses, we used the CDOCKER Interaction Energy scoring function and MMGBSA binding energy, as during the previous validation [26] they had achieved the best Pearson correlation coefficients (0.62 and -0.66, respectively) with the pIC50 values of compounds tested in vitro for PLpro inhibition. For 7jn2-derived mutations, we used

CDOCKER Interaction Energy and Jain scoring functions, and MMGBSA binding energy (R = 0.56, 0.73 and -0.64, respectively). Here, we conducted analogical validation for 7d7t and chose CDOCKER Interaction Energy and PMF04 scoring functions (R = 0.56 and 0.65, respectively).

## Molecular dynamics simulations

All the MD simulations were done using AMBER18 package with ff14SB force field. Starting structures were obtained from Protein Data Bank (PDB)—PLpro-GRL-0617 complex (PDB ID: 7jrn) and PLpro-S43 complex (PDB ID: 7d7t; during preparation of this manuscript new structure superseded 7d7t—7e35; superposition between these two structures is shown in S2 Fig). Ligand parametrization was done using ANTECHAMBER and GAFF. For maintaining proper interaction between zinc ion and cysteines (C189, C192, C224, C226) Zinc AMBER force field (ZAFF) was used [45]. The complexes were solvated using TIP3P water and $Na^+$ and $Cl^-$ were added to neutralize the charge and obtain 0.1 M of ionic strength. An integration step of 2 fs was used and all bonds involving hydrogens were constrained with SHAKE algorithm. Cutoff of 10 Å was used for electrostatic interactions. During the first phase of equilibration Langevin thermostat was set to gradually increase temperature of the system from 0 to 310 K for 500 ps and then hold the set temperature for the next 500 ps. In the next step, we added pressure control using Berendsen barostat set for 1 atm and 1 ns. During the next four equilibration steps (each lasting 1 ns) we were gradually releasing the constraints and lowering its force (starting from frozen backbone of the protein and all heavy atoms of the ligand with 100 kcal/mol/Å$^2$ force to only $C_\alpha$ atoms constrained with 0.1 kcal/mol/Å$^2$). The production was set to 500 ns. In some of the trajectories the ligand dissociated from the binding site, thus for the analysis we used fragments of the trajectories with ligand still bound to the site (all the trajectories we performed along with their length are presented in S1 Table). The mutations were done in PyMol. Ligand clustering was done using hierarchical clustering in AMBER software and cutoff 1.5 Å. For the estimation of protein-ligand binding energy we used frames every 100 ps and MMPBSA.py from AMBER software. We performed MMGBSA calculation using igb = 5 and salt concentration of 0.1 M. The binding free energy was calculated using following formulas: $\Delta G_{binding} = G_{complex} - G_{protein} - G_{ligand}$; $G = \Delta G_{el} + \Delta G_{nonel} + \Delta E_{MM}$, where $\Delta E_{MM}$ is the interaction energy in gas phase, and $\Delta G_{el}$ and $\Delta G_{nonel}$ correspond to polar and nonpolar components. The detailed formulas for the specific terms are available in the freely accessible Amber 2018 Reference Manual. Note, that the entropic contribution was not calculated because of high computational costs involved with the considerable number of frames we performed the calculation for. The averages and standard errors shown on Fig 3D are calculated using block averages. For each trajectory we obtained the standard error of the average based on trajectory segments of length $n$ (blocks). For each $n$ we calculated first the standard deviation among the block averages ($\sigma_n$) and then the standard error: $\sigma_n/\sqrt{M}$, where M is number of blocks of length $n$. Then, for each trajectory we chose $n$ based on the plateau of the standard error.

## Experimental procedures

**Plasmids.** The pET21a plasmid with wild type PLpro protein sequence with uncleavable C-terminal 6xHis tag was a kind gift from Dr. Olsen from Department of Biochemistry and Structural Biology University of Texas Health Science Center at San Antonio, San Antonio, TX 78229, USA. All PLpro variants P247S, P247L, P247T, P248S, P247S-P248S, E263D, Y264H and E263D-Y264H were prepared by site-directed mutagenesis (QuickChange, Stratagene).

**Expression and purification of recombinant wild type PLpro and its variants.** The wild type SARS-CoV-2 PLpro protein and its variants were expressed and purified as previously described [43]. The purity of the collected fractions was analyzed by SDS-PAGE. Then, purified proteins were desalted on PD-10 columns and frozen at -80˚C for later use.

**Circular dichroism spectroscopy.** CD spectra were acquired for 5–6 µM protein concentration in 5 mM Tris buffer pH 7.5 at 20˚C in 200–260 nm range on JASCO-1500 CD spectropolarimeter. The width of the cuvette was 0.1 mm.

**Mutant's activity assessment.** To determine activity of the recombinant SARS-CoV-2 PLpro mutants, each mutant was preincubated in the assay buffer (pH 7.5, Tris 50 mM, NaCl 5 mM, DTT 2 mM, 0.075% BSA) for 30' at 37˚C. Then, 99 $\mu$L of the enzyme was added to the wells of 96-well plate containing 1 $\mu$L of substrate (Ac-LRGG-ACC DMSO solution). Final concentrations during the measurements: [E]=100 nM, [S]=10 $\mu$M. ACC fluorophore liberation was measured using SoftMax software with SpectraMax Gemini XPS microplate reader at $\lambda$ex = 355 nm and $\lambda$em = 460 nm for 45'. For the analysis the linear portion of the progress curved was used. The experiment was repeated at least two times.

**IC50 determination.** To determine IC50 for the S43 and GRL-0617 inhibitors, SARS-CoV-2 PLpro mutants were preincubated in the assay buffer for 10' at 37˚C. Then, 79 $\mu$L of the enzymes was added to the wells of 96-well plate containing 1 $\mu$L of the inhibitor DMSO solutions in wells. Next, enzymes were incubated with inhibitors for 30' at 37˚C. After the incubation 20 $\mu$L of the substrate in the assay buffer was added to the wells. Final concentrations during the measurements: [E]=100 nM, [S]=10 $\mu$M, inhibitors were diluted in DMSO at 14 concentrations ranging from 100 $\mu$M to 12 nM (1/2 serial dilutions). ACC fluorophore liberation was measured using SoftMax software with SpectraMax Gemini XPS microplate reader at $\lambda$ex = 355 nm and $\lambda$em = 460 nm for 45'. For the analysis the linear portion of the progress curved was used. The experiment was repeated three times.

## Supporting information

**S1 Fig. PLpro sequences by month.** Lines show distribution of the sequences carrying the most frequent variants. For example, in April and May 2021 more than 60% of the mutated sequences had a modification on 145 position. Shaded areas show number of deposited PLpro sequences— light green show all of the sequences, dark green only those carrying at least one mutation. From April to June 2021 the majority of the deposited sequences had at least one variant.
(PDF)

**S2 Fig. Superposition of the PLpro-S43 structures.** Based on PDB IDs 7d7t (red; removed from PDB) and 7e35 (blue; supersedes 7d7t). $C_\alpha$ RMSD is 0.4 Å, RMSD of all heavy atoms is 0.8 Å.
(TIF)

**S3 Fig. $C_\alpha$ RMSD of the PLpro-ligand trajectories.** Left: GRL-0617, right: S43. The trajectories in which the ligand dissociated from the binding site are marked with red—the length of these trajectories is truncated to the time the ligand stayed in the site (see S1 Table).
(PDF)

**S4 Fig. GRL-0617 interaction energy calculated using PLpro-ligand trajectories.** The calculations are based on MMGBSA method. The trajectories in which the ligand dissociated from the binding site are marked with red—the length of these trajectories is truncated to the time the ligand stayed in the site (see S1 Table).
(TIF)

**S5 Fig. Per-residue decomposition interaction energy calculated using PLpro-ligand trajectories.** The calculations are based on MMGBSA method.
(TIF)

**S6 Fig. Selected variants found in PLpro.** Left: Location of amino acids subject to mutation relative to the ligand (GRL-0617) binding site (based on PDB ID: 7jrn). Right: Representative ligand conformations bound to PLpro. Based on clustering of ligand conformations from Molecular Dynamics trajectories (sampled every 100 ps).
(TIF)

**S7 Fig. Evaluation of secondary structure of recombinant SARS-CoV-2 PLpro variants.** Circular dichroism (CD) measurements for wild type of PLpro protein and its variants displayed a spectrum which shows negative ellipticity between 205 and 240 nm. It suggests that all of them may be in native state and have similar to wild type scaffold.
(TIF)

**S1 Table. Length of the PLpro-inhibitor trajectories (in ns) analyzed in this study.** The bolded entries with asterisk indicate the trajectories in which the ligand dissociated from the binding site. For analysis, only the frames with ligand present in the binding site were used.
(PDF)

**S2 Table. Hydrophobic interactions between S43 ligand rings and protein amino acids.** Crystal—data for PDB ID: 7d7t with information on distance between rings. For trajectories—percentage of trajectory in which interaction occurs (i.e. D < 5Å; based on frames every 100ps). Trajectories in which the ligand dissociated from the protein are indicated with *. A1, A2—naphthyl group; B—piperidine; C—benzene.
(PDF)

**S3 Table. Hydrophobic interactions between GRL-0617 ligand rings and protein amino acids.** Crystal—data for PDB ID: 7jrn with information on distance between rings. For trajectories—percentage of trajectory in which interaction occurs (i.e. D < 5Å; based on frames every 100ps). Trajectories in which the ligand dissociated from the protein are indicated with *. A1, A2—naphthyl group; B—4-methylbenzenamine moiety.
(PDF)

**S1 File. All variants of PLpro.** Number of sequences with each variant we found.
(XLSX)

**S2 File. Mutation and protein-ligand binding energies.** Calculations based on three PLpro-inhibitor complexes (PDB ID: 7jn2, 7jrn and 7d7t).
(XLSX)

## Author Contributions

**Conceptualization:** Alicja W. Maksymiuk, Marcin Drag, Joanna I. Sulkowska.

**Data curation:** Karolina Swiderska, Mikolaj Zmudzinski, Joanna I. Sulkowska.

**Formal analysis:** Agata P. Perlinska, Adam Stasiulewicz, Mai Lan Nguyen, Karolina Swiderska, Alicja W. Maksymiuk.

**Funding acquisition:** Marcin Drag, Joanna I. Sulkowska.

**Investigation:** Agata P. Perlinska, Adam Stasiulewicz, Mai Lan Nguyen, Karolina Swiderska, Mikolaj Zmudzinski, Alicja W. Maksymiuk, Joanna I. Sulkowska.

**Methodology:** Agata P. Perlinska, Adam Stasiulewicz, Mai Lan Nguyen, Karolina Swiderska, Mikolaj Zmudzinski, Joanna I. Sulkowska.

**Project administration:** Joanna I. Sulkowska.

**Resources:** Joanna I. Sulkowska.

**Supervision:** Marcin Drag, Joanna I. Sulkowska.

**Validation:** Agata P. Perlinska, Adam Stasiulewicz, Alicja W. Maksymiuk, Joanna I. Sulkowska.

**Visualization:** Agata P. Perlinska, Alicja W. Maksymiuk, Joanna I. Sulkowska.

**Writing – original draft:** Agata P. Perlinska, Adam Stasiulewicz, Mai Lan Nguyen, Alicja W. Maksymiuk.

**Writing – review & editing:** Agata P. Perlinska, Adam Stasiulewicz, Mai Lan Nguyen, Marcin Drag, Joanna I. Sulkowska.

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
