## [Decision Letter · Decision Letter 0]

5 May 2022

Dear DR Sulkowska,

Thank you very much for submitting your manuscript "Amino acid variants of SARS-CoV-2 papain-like protease have impact on drug binding" for consideration at PLOS Computational Biology.

As with all papers reviewed by the journal, your manuscript was reviewed by members of the editorial board and by several independent reviewers. In light of the reviews (below this email), we would like to invite the resubmission of a significantly-revised version that takes into account the reviewers' comments.

We cannot make any decision about publication until we have seen the revised manuscript and your response to the reviewers' comments. Your revised manuscript is also likely to be sent to reviewers for further evaluation.

Sincerely,

Alexander MacKerell

Associate Editor

PLOS Computational Biology

Nir Ben-Tal

Deputy Editor

PLOS Computational Biology

Reviewer's Responses to Questions

**Comments to the Authors:**

Reviewer #1: The manuscript 'Amino acid variants of SARS-CoV-2 papain-like protease have impact on drug binding' by Perlinska et al.

presents a study on the effect of mutations on the binding mode of potential inhibitors to the Papain-like Protease (PLP)

of SARS-CoV2. The study combines Molecular Dynamics (MD) simulations with molecular docking, sequence analysis, MMPBSA

calculations and experimental enzyme activity assays to characterize the impact of five mutations : P247S, E263D-Y264H and

T265A-Y268C on the affinity of potential non-covalent inhibitors of PLP.

The manuscript is well-structured and all methods that have been used in this study are described in detail.

The authors' conclusions are supported by the results, although this referee has some remarks on the methodology

used in their study (see below). This referee recommends this manuscript to be published after a minor revision.

In the following, I will point out my remarks that appeared to me during the study of this manuscript :

1. Sequence redundancy in the selected dataset :

The selected data of PLP-sequences seems to be highly redundant, which means that the sequence-similarity

among a large fraction of the selected CoV-2 datasets seems to be redundant as well. To remove artefacts that

arise from sequence-redundancies, only sequences with a similarity that lies below a certain threshold should

be considered < 80 %. (see the mutation rates in Table 1).

The mutation-rates can be reflected using a position-dependent heat-mapped color-coding on Figure 1.

2. Median of MMPBSA-interaction energies :

The median values and the variances of the measured energies (see Figure 3 D) indicate that there is no

evident difference in the energies between the mutational variants and the wild-type, because the error-bars

are crossing each of the individual median values.

A running averaging that also considers the statistical error might be more indicative for the energy

differences : 100 ps, -> av_2 1 ns -> av_3 10 ns -> av_4 100 ns -> av_5 500 ns (final result) + the statistical error.

As MMPBSA is a quite inaccurate method for the calculation of interaction energies, the study could be improved

using thermodynamic integration (TI) or Free energy perturbation (FEP) calculations.

3. The RMSDs of the residues in the drug-binding pocket over time can be added for each mutational variant, as this metric might

be an indicator for the stability of the drug-binding site compared to the wild-type. Further, it might explain the

differences in the enzymatic activity that has been observed experimentally.

Reviewer #2: In this manuscript, the authors tried to clarify different sequences of PLpro and then studied the influence of these mutations on the binding process of ligands to PLpro via docking and MD simulations. In vitro works were then performed to validate the observation. It is of great interest to read the manuscript. A large work was completed, however, there are some comments to improve the manuscripts.

1. The structural change of the PLpro under the effect of mutation would significantly impact the binding free energy and binding pose of ligands to PLpro. So, the MM/GBSA calculation should be carried out over the equilibrium snapshots of the complex, which were obtained from MD simulations instead of molecular docking only.

2. According to the previous comment, the MM/GBSA calculation over docking simulation probably does not make sense since the obtained results are not significantly different eg. the binding energy range from -27.4 +/- 6.1 to -32.28 +/- 5.7 (line 286), or -28.1 +/- 3.6 to -29.0 +/- 3.5 kcal/mol (line 314 - 315), etc. The obtained results are not different within the error bar, authors may wish to discuss about this.

3. The MM/GB(PB)SA calculation (Per-residue free energy decomposition) should be carried out since the obtained results will clarify the contribution of each residue of PLpro. The interaction picture would thus be a clarifier.

4. The MMGBSA terms should be reported in the manuscript.

**Have the authors made all data and (if applicable) computational code underlying the findings in their manuscript fully available?**

Reviewer #1: Yes

Reviewer #2: Yes

PLOS authors have the option to publish the peer review history of their article (what does this mean?). If published, this will include your full peer review and any attached files.

Reviewer #1: No

Reviewer #2: No
---

## [Decision Letter · Decision Letter 1]

19 Oct 2022

Dear DR Sulkowska,

We are pleased to inform you that your manuscript 'Amino acid variants of SARS-CoV-2 papain-like protease have impact on drug binding' has been provisionally accepted for publication in PLOS Computational Biology.

Best regards,

Alexander MacKerell

Academic Editor

PLOS Computational Biology

Nir Ben-Tal

Section Editor

PLOS Computational Biology

Reviewer's Responses to Questions

**Comments to the Authors:**

Reviewer #1: This referee thinks that the article is suitable for publication.

Reviewer #2: Authors answer all of my question. The manuscript should be published at the current form.

**Have the authors made all data and (if applicable) computational code underlying the findings in their manuscript fully available?**

Reviewer #1: Yes

Reviewer #2: **No: **

PLOS authors have the option to publish the peer review history of their article (what does this mean?). If published, this will include your full peer review and any attached files.

Reviewer #1: No

Reviewer #2: No

---

## [Editor Report · Acceptance letter]

26 Oct 2022

PCOMPBIOL-D-22-00373R1 

Amino acid variants of SARS-CoV-2 papain-like protease have impact on drug binding

Dear Dr Sulkowska,

I am pleased to inform you that your manuscript has been formally accepted for publication in PLOS Computational Biology. Your manuscript is now with our production department and you will be notified of the publication date in due course.

With kind regards,

Zsofia Freund
